# Transcription Factor Zbtb20 as a Regulator of Malignancy and Its Practical Applications

**DOI:** 10.3390/ijms241813763

**Published:** 2023-09-06

**Authors:** Dimo Stoyanov, George S. Stoyanov, Martin N. Ivanov, Radoslav H. Spasov, Anton B. Tonchev

**Affiliations:** 1Department of Anatomy and Cell Biology, Medical University of Varna, 9000 Varna, Bulgaria; 2Department of Clinical Pathology, Complex Oncology Center, 9700 Shumen, Bulgaria; 3Department of Stem Cell Biology, Research Institute, Medical University of Varna, 9000 Varna, Bulgaria

**Keywords:** Zbtb20, malignant tumors, hepatocellular carcinoma

## Abstract

Zbtb20 (zinc finger and BTB domain-containing protein 20) is a transcription factor with a zinc finger DNA binding domain and a BTB domain responsible for protein–protein interaction. Recently, this TF has received attention because new data showed its pivotal involvement in normal neural development and its regulatory effects on proliferation and differentiation in different tissues. Zbtb20 was shown to increase proliferation and migration and confer resistance to apoptosis in the contexts of many malignant tumors like hepatocellular carcinoma, non-small-cell lung carcinoma, gastric adenocarcinoma, glioblastoma multiforme, breast cancer, and acute myeloid leukemia. The involvement of Zbtb20 in tumor biology is best studied in hepatocellular carcinoma, where it is a promising candidate as an immunohistochemical tumor marker or may be used in patient screening. Here we review the current data connecting Zbtb20 with malignant tumors.

## 1. Introduction

Zinc fingers are a large group of proteins that contain at least one of the following domains along with the zinc fingers: KRAB, BTB, SET, and SCAN [1]. The importance of zinc finger proteins in cancer biology was excellently reviewed by Zhao et al. [2]. Several members of the ZBTB protein group are well established in the field of cancer biology both as tumor suppressors, such as ZBTB16 and ZBTB28, and oncogenes like ZBTB3, ZBTB7A, ZBTB38, and others [3,4,5,6,7,8]. The zinc finger and BTB domain-containing 20 (Zbtb20) gene encodes for a transcription factor (TF), known for its involvement in both pre- and postpartum development as well as in pathological conditions, ranging from inflammation and tissue repair to oncogenesis. The BTB domain of Zbtb20 is responsible for protein–protein interaction, allowing for the formation of dimers, while the five zinc fingers allow for specific DNA binding, necessary to regulate gene expression [9,10]. The gene is located on the first outermost band of the long arm of chromosome 3 (3q13.31) [11]. The protein shows a predominantly nuclear expression pattern [12,13,14,15]. It also has two isoforms—a long one made up of 741 amino acids, and a short one. The short isoform has its N-terminus truncated and is made up of 668 amino acids in total but has all the domains contained in the long form (Figure 1) [9]. Both appear to be largely similar in function [9,16]. The earliest reported Zbtb20 expression in humans was in dendritic cells [9]. Later, its presence was identified in the central nervous tissue. Thus far, its function in hippocampal development is best understood, since Zbtb20 mutants have easily recognizable hippocampal defects [9,12,17]. Subsequent reports revealed the importance of Zbtb20 in the normal development of the neocortex, olfactory bulb, hypophysis, and skeletal system [13,15,18,19]. The second most discussed organ is the liver, due to the metabolic problems that are present in mutant animals—impaired glucose homeostasis and lipid metabolism. A lot of attention has been focused on the involvement of Zbtb20 in cell cycle control and the modulation of the inflammatory response. After hepatic lobectomy, Zbtb20 knockout mice showed a significant decrease in hepatocyte proliferation in the injured liver, gradually returning to the control levels. This was primarily due to a delay in the induction of EGFR, which also reaches its physiological benchmarks [20]. After Zbtb20 knockout, lower cell division rates were also reported in chondrocytes, cortical radial glia progenitors, and post-stroke adult astrocytes [13,15,18]. Conversely, in B-lymphocytes, Zbtb20 acted like a cell cycle repressor: the overexpression of this TF led to a halt in the cell cycle in the G1 phase [21]. A regulatory role in the cell cycle of Zbtb20 is conceptually consistent with a role in tumorigenesis. Recently, it was found that HBV can integrate into the Zbtb20 gene, and an association was observed between the integration and Zbtb20 expression in patients with hepatocellular carcinoma (HCC) [22]. The experiments in the liver placed Zbtb20 in the spotlight and opened the way to more research to better understand its function in tumorigenesis. According to the literature, it is currently implied in the tumor biology of HCC, non-small-cell lung carcinoma (NSCLC), gastric adenocarcinoma (GAC), glioblastoma multiforme (GBM), breast cancer, and acute myeloid leukemia (AML). Here we aim to summarize the current data that explain the function of this TF in oncogenesis.

## 2. Zbtb20 and Malignant Tumors

Typical features of malignant tumors are (1) limitless and self-sufficient proliferation and resistance to growth inhibition; (2) evasion of apoptosis; (3) increased cell motility and tissue invasion; and (4) neoangiogenesis. The modulatory effects of Zbtb20 on these features are summarized in Figure 2.

### 2.1. Limitless and Self-Sufficient Proliferation, and Resistance to Growth Inhibition

Two independent research teams have reported that FOXO1, a member of the fork-head box (FOX) TF family and a known suppressor of cell division, is repressed by Zbtb20, which directly binds to its regulatory region in both HCC and NSCLC. Overexpression of Zbtb20 suppressed FOXO1 and promoted proliferation [23,24]. It is worth mentioning that Zbtb20^−/−^ mice showed slower proliferation during liver regeneration after liver lobectomy, measured by BrdU uptake [20]. Furthermore, increased colony formation capabilities were reported during Zbtb20 overexpression in HCC cell cultures [23]. Higher protein and mRNA levels of Zbtb20 due to either forced Zbtb20 expression or to the silencing of Zbtb20 mRNA-interacting microRNAs (miR) were concomitant with increased proliferation in HCC, NSCLC, GBM, GAC, AML, and breast cancer. By contrast, decreased expression of the Zbtb20 protein had the opposite effects [23,25,26,27,28,29].

### 2.2. Evasion of Apoptosis

Several different pathways can trigger programmed cell death (apoptosis). One experimental approach to test apoptotic resistance is to introduce chemical substances that can induce DNA damage to cell cultures. A recent study took advantage of this approach using damage-inducing drugs such as etoposide or adriamycin. They found an increase in resistance to apoptosis of plasma cell cultures where Zbtb20 is overexpressed [21]. Similarly, passive apoptotic levels without induction were comparable in breast cancer, AML, and GBM cell cultures [25,28,30,31]. When Zbtb20 was knocked down by the overexpression of miR-758-5p in transfected GMB cell cultures, the number of apoptotic cells significantly increased by approximately 40%. Thus, Zbtb20 seems to help cells evade physiological apoptotic responses and speed up cancer propagation. Regardless of the established anti-apoptotic properties of Zbtb20, the molecular foundations of this effect are still unknown.

### 2.3. Increased Cell Motility Leading to Tissue Invasion

Metastatic tumor cells can migrate through the native tissue, invade nearby blood or lymphatic vessels, survive in the circulatory system, invade vessels at a distant location, and reproduce themselves in this new environment. For migration to occur, several criteria need to be met: first, cell-to-cell adhesion molecules should be down-regulated; second, migratory pathways should be activated, allowing for cytoskeleton remodeling; third, digestion of the basal membranes should be achieved. The migration of human cancer cell lines has been assessed in conditions where Zbtb20 was overexpressed or silenced. Both transwell and wound scratch assays showed higher migratory capabilities when Zbtb20 was overexpressed, compared to the control cell lines [32]. As expected, silencing experiments had the opposite effects on migration [32]. This higher migratory phenotype was concomitant with increased activity and expression of matrix metalloprotease (MMP) 2 and 9 in GAC [32]. Migration was also promoted in cell lines from GBM, breast cancer, and AML during transwell assays [25,28,29].

### 2.4. Chronic Inflammation

Chronic inflammatory response carries a significant risk for tumor formation by enhancing cell proliferation and promoting transient cell metaplasia [33,34]. Chronic inflammation generates reactive oxygen species and reactive nitrous species, which can lead to DNA damage [35]. This mutagenic effect, combined with the pro-proliferative signals and metaplastic processes over a prolonged period, can lead to tumorigenesis. Treatment of NSCLC cells with the pro-inflammatory cytokines tumor necrosis factor-alpha (TNF-α) and interleukin 1b (IL1b) increased the expression of Zbtb20 [26]. Helicobacter pylori (H. pylori) plays an essential role in the pathogenesis of GAC [36]. Since gastric epithelial cells express pattern recognition receptors like toll-like receptors, they can be induced to secrete interleukins (IL-8, TNF-α, and IL-1β) as well as reactive oxygen species in response to bacterial infection [37]. Infecting GAC cell cultures with H. pylori increased Zbtb20 expression. Furthermore, the inflammatory nuclear factor kappa-light-chain-enhancer of activated B cells (NF-κB) pathway was promoted after H. pylori infection [32,33]. It is possible that these effects are due to direct interactions between H. pylori and the gastric epithelia by some signaling molecules like micro RNAs or pattern recognition receptor activation. In any case, the end effect is the activation of the anti-inflammatory TF Inhibitor of NF-κB α (IκBα). These two studies give us a glimpse of a possible pathway in the pathogenesis of malignant tumors where Zbtb20 expression is modulated by chronic inflammatory stress. Significantly more data on the regulation of NF-kB are available from non-cancer studies, which will be discussed below.

### 2.5. Anti-Tumor Immunity

Immune cells are a key feature of the tumor microenvironment. A recent study provided a glimpse into the role of Zbtb20 in tumor immunity and showed that deleting Zbtb20 enhances the anti-tumor activity of CD8+ cytotoxic T cells [38]. This is quite interesting since Zbtb20 overexpression is usually linked to a more malignant tumor phenotype.

### 2.6. Neoangiogenesis

The formation of new blood vessels is vital for tumor growth since diffusion allows for lesions no bigger than 2 mm. Angiogenesis also facilitates metastasis. At present, there are no published data on blood vessel growth and Zbtb20 expression in tumors.

## 3. Molecular Partners of Zbtb20

So far, we presented the effect of Zbtb20 on cell biology in the context of a tumor environment. Some of the molecular mechanisms behind its effects are already known. Most of the implicated signaling pathways like epidermal growth factor receptor (EGFR), WNT, and FOXO1 are important for cell proliferation, adhesion, and migration. Furthermore, Zbtb20 is regulated by interactions with a myriad of long non-coding RNAs (lncRNAs), circular RNAs, and miRs, so direct mutation to the gene could be absent in the tumor tissue, but its expression can still be dysregulated. In Figure 3, we aim to summarize the known interactions of Zbtb20 with other molecules in the context of tumorigenesis.

### 3.1. Regulation of Cell Cycle by FOXO1 and Downstream Effectors

FOXO1 is a known cell cycle repressor and has an inverse correlation with Zbtb20 levels. Two independent studies identified the promoter region of FOXO1 as a direct binding site for Zbtb20, which induces gene repression [23,26]. FOXO1 inhibits CyclinD synthesis, required for cell cycle progression while activating two proteins impinging on the cell cycle—cyclin-dependent kinase inhibitors p21/p27 (CKIs p21/p27). This interaction allows Zbtb20 to influence proliferation through the FOXO1 signaling pathway [24,39]. Consistent with these data, CyclinD/CyclinE were up-regulated, while CKIs p21/p27 were repressed when Zbtb20 was overexpressed in HCC and NSCLC cultures. This was accompanied by lower expression of FOXO1 [23,26]. Additionally, in model animals subjected to liver lobectomy, the Zbtb20^−/−^ mutant group showed perturbed induction of CyclinD1 [20].

### 3.2. Liver Regeneration and EGFR Induction

Mice, subjected to liver lobectomy, exhibited delayed regeneration [20]. Hepatocyte priming by interleukins and TNF-α was normal, but EGFR protein and mRNA levels were substantially decreased. Constitutive overexpression of Zbtb20 by viral vectors partially restored EGFR levels. Interestingly, chromatin immunoprecipitation assays showed that Zbtb20 did not directly regulate EGFR expression, as no binding to the EGFR gene was observed [20]. This matches well with the fact that the entire group of Zbtb proteins is considered to function mainly as transcriptional repressors. There is direct evidence of such a function of Zbtb20 on some genes [9,14,23,26,40]. In this context, it is quite possible that Zbtb20 could activate EGFR transcription indirectly, possibly by inhibiting some of its repressors. Further research will be needed to shed more light on the Zbtb20/EGFR pathway.

### 3.3. WNT/b-Catenin Pathway

The canonical and noncanonical Wnt signaling pathways are responsible for cell division and cell migration/adhesion, respectively. Both processes are essential in tumor physiology. Upon activation, they lead to a reduction in the so-called β-catenin destruction complex (APC/Axin/GSK3), which phosphorylates β-catenin and targets it for proteasomal degradation [41]. The accumulated β-catenin is translocated to the nucleus and binds to TFs like LEF/TCF, leading to the activation of downstream genes [41]. Overexpression of Zbtb20 by the piggyBac transposon system led to an increase in the phosphorylation of β-catenin, along with an increase in the levels of AXIN1, a downstream molecule in the pathway. This was concomitant with a decrease in the expression of members of the destruction complex, like GSK3B and APC [16]. The interactions between Zbtb20 and the WNT pathways are most likely mediated, in some way, by PPARG, as PPARG levels are significantly lower when Zbtb20 is overexpressed [16]. Furthermore, when the Zbtb20 function was disrupted by CRISPR/CAS9 mutation, the expression of downstream targets like Cyclin D1 and Myc was much lower. According to the authors, the Omnibus database shows upregulation in Zbtb20 expression in all stages of HCC, accompanied by WNT signaling overactivation [16]. As we mentioned above, liver regeneration is accompanied by EGFR induction, which is perturbed in Zbtb20^−/−^ animals, although the molecular mechanisms of this are unknown. Interestingly, both the EGFR and WNT pathways are often simultaneously overreactive in tumor tissues [42]. This led to the idea of crosstalk between the two pathways. For example, overstimulation of the WNT pathway leads to MMP synthesis, which cleaves and releases EGFR ligands [42]. Since Zbtb20 is implicated in both pathways, it is possible to at least partially mediate this “crosstalk”.

### 3.4. AFP (Alpha-Fetoprotein) Repression

AFP is a globulin family protein expressed during human development with an uncertain function, although some evidence suggests it is involved in the transport of metal ions and non-soluble molecules [43]. It is increased in patients with HCC and is an established biomarker in its diagnosis [44,45]. One should interpret AFP levels with caution and never solely rely on this one marker for diagnosis. It is also increased in other malignant tumors as well as in non-neoplastic liver pathologies [45]. In the liver, Zbtb20 is a direct repressor of Afp [40]. Conversely, HCC patients showed no correlation between serum AFP and Zbtb20 levels. This discrepancy was addressed by some AFP immunohistochemical studies pointing out that the tissue but not serum levels of AFP inversely correlated with Zbtb20 expression [46].

### 3.5. Response to Inflammation and Migration

As mentioned above, Zbtb20 expression is associated with the inflammatory pathway NF-κB/IκBa. NF-κB is a TF that is sequestered in the cytoplasm when bound by its inhibitor IκBa. The canonical activation of this pathway requires the phosphorylation of IκBa, tagging it for proteasomal degradation, reducing its cytoplasmic levels, and thus allowing for the nuclear translocation of NF-κB. Overexpression of Zbtb20 reduces IκBα levels, increasing those of phosphorylated NF-κB [32]. Phosphorylation of NF-κB has variable effects on its activity and can both promote or inhibit this regulatory pathway, although activation seems more likely in the context of this experiment [47,48]. Nevertheless, the substantial reduction in the IκBα level after Zbtb20 stimulation indicates pathway activation. To our knowledge, there are no data on how these genetic regulations are achieved. MMP2/9 levels were also elevated upon Zbtb20 overexpression in the same GAC cell cultures [32]. An increase in MMP2/9 was also reported in breast cancer cell cultures [29]. This effect is probably achieved via the NF-κB pathway, although a direct effect of Zbtb20 on MMP2/9 cannot be excluded [49,50]. Interestingly a separate study pointed out that Zbtb20 gene polymorphism was associated with increased risk of GAC and esophageal carcinoma [51]. The gene is also implicated in GAC by genome-wide association studies [52].

### 3.6. Interaction with Micro-RNAs, Long Non-Coding RNAs, and Circular RNAs

Gene expression can be modulated on multiple levels. A large and diverse group of molecules that are responsible for regulating gene expression on the post-translational level, is the miRs—small RNA molecules with a length of around 22 nucleotides. They form a silencing complex with the endonuclease AGO and can bind the so-called miR response elements, usually located on the 3′ untranslated region (3′ UTR) of mRNAs, effectively inhibiting protein synthesis [53]. miR can also modulate gene transcription by binding to the promoter or 5′ UTR region. LncRNAs are RNA molecules longer than 200 nucleotides, which can also regulate gene transcription. Some LncRNAs can sequester miRs by complementary binding to them [54]. This process is called miR “sponging” [54]. Another class of RNA sponges are circular RNAs—non-coding RNAs that are generated by 3′-5′ covalent binding [55]. Several miRs have been shown to bind Zbtb20 mRNA 3′ UTR. miR 758-5p synthesis is repressed in GBM and correlates with a malignant phenotype. Similarly, lower levels of miR-378a were reported in AML, as compared to healthy subjects (due to LncRNA-00641 sponging). In both cases, overexpressing these miRs leads to decreased Zbtb20 protein levels and lower proliferation of the respective tumor cell cultures [25,28]. In breast cancer, the miR-634 binds to the 3′ UTR of Zbtb20 and lowers its mRNA levels. The LncRNA SNHG8, which shows low expression in tumor tissues, represses the synthesis of miR-634 and thus disinhibits the Zbtb20 signaling pathway [29,56]. Breast cancer progression is also increased by the action of a circular RNA—circ-0104345. It regulates Zbtb20 expression by releasing the gene from the inhibitory action of miR-876-3p [31]. Zbtb20 was regulated in a similar fashion by another pair of RNAs—circ-SFMBT2 and miR-582-3p in AML cell lines [30]. MiR-122 and miR-214 also modulate Zbtb20, as miR-122 is associated with higher Zbtb20 levels and poor patient prognosis [57]. The exact regulatory mechanism will be discussed below. In summary, different non-coding RNA types can regulate Zbtb20 expression, either directly or indirectly. Regardless of the exact mechanisms, in all the above-mentioned cases, the dysregulation of the non-coding RNAs that modulate Zbtb20 expression increases Zbtb20 synthesis and leads to malignant tumor growth.

### 3.7. Regulation by CUX1

MiR-122 is highly expressed in hepatocytes, accounting for around 70% of all miRs in normal liver tissue, while in HCC the levels of this miR are severely decreased [58]. MiR-122 binds to the mRNA of the TF CUX1 (cut homeobox). CUX1 overexpression is associated with a poor prognosis in patients with HCC. Silencing of the miR-122 leads to overexpression of CUX1 and concomitant repression of Zbtb20. This effect is achieved by the direct promoter activity of CUX1 on the transcription of miR-214, which in turn binds to the Zbtb20 mRNA, thus inhibiting its translation [57]. Low serum levels of AFP in HCC patients correlate with positive CUX1 status [57]. This pathway suggests that at least part of the HCC, characterized by low levels of Zbtb20, could be CUX1-positive.

## 4. Lessons from Other Conditions

### 4.1. Further Evidence for the Regulation of the EGFR Pathway

The role of Zbtb20 in EGFR activation, first reported in a liver regeneration study, was further supported by research on cardiac hypertrophy [59,60]. The authors found a stronger hypertrophic response to angiotensin II treatment in Zbtb20 overexpressing mice, which was accompanied by an increase in the levels of molecular markers for adverse cardiac remodeling like atrial natriuretic peptide, brain natriuretic peptide, and the β-myosin heavy chain [59]. This phenotype was accompanied by stronger activity of the EGFR promoter region, measured by luciferase activity. Silencing Zbtb20 had the opposite effect [59].

### 4.2. Further Evidence for the Regulation of Inflammation and the NF-kB Pathway

We already discussed the effect of Zbtb20 on the NF-kB pathway in CAG. Zbtb20 was also shown to modulate the NF-kB pathway in different settings: again in cardiac tissue after transplantation, in ox-LDL stimulated macrophages, in bone tissue, and in myeloid macrophages [61]. After heart tissue transplantation, silencing Zbtb20 expression alleviated the rejection of cardiac allografts. This was accompanied by an increase in the anti-inflammatory M2 macrophages (M2 macrophage polarization). The Zbtb20 knockdown animals had lower levels of the phosphorylated p-65 (one of the active forms of NF-kB) and less invasion of inflammatory cells like the CD3+ T-lymphocytes. Adding betulinic acid (an NF-kB activator) raised the phosphorylated p-65 to control levels and increased the inflammatory cell invasion of the allografts [61]. In a separate study, Zbtb20 was identified as a hub gene for inflammation and oxidative stress in ox-LDL-stimulated macrophages. Like in the previous study, Zbtb20 knockdown reduced the levels of phosphorylated p-65. Furthermore, the authors showed a reduction in the levels of the proinflammatory cytokines TNF-α and IL-6. Additionally, players in the MAPK pathway, like ERK and JNK, were also reduced [62]. Similar effects were reported in bone-marrow-derived macrophages, stimulated by titanium particles. The particles induced the standard NF-kB response: more phosphorylated forms of pathway proteins like p-65 and IkBα, increased IL synthesis, and M1 macrophage polarization, all of which are attributed to Zbtb20. The authors suggest that Zbtb20 may directly bind to IkBα and repress the promotor activity, indicated by a luciferase assay [63]. Data derived from myeloid-specific Zbtb20^−/−^ knockouts showed by means of ChiP and high-throughput next-generation sequencing that Zbtb20 specifically binds the promotor of IkBα and thus activates the NF-kB upon TLR (toll-like receptor) stimulation [64]. The myeloid Zbtb20^−/−^ had lower activation of the inflammatory response to lipopolysaccharide (a TLR ligand) treatment, with lower levels of IL synthesis.

### 4.3. Further Evidence on the Evasion of Apoptosis

Evasion of apoptosis was also reported in nucleus pulposus cell lines. Overexpression of the Lnc RNA KLF3-AS1 increased the expression of Zbtb20 and resulted in increased cell viability and reduced apoptosis while decreasing Zbtb20 levels, although miR-10a-3p overexpression had the opposite effect [65].

## 5. A Practical Approach to Zbtb20 Tissue Expression in Malignant Tumors

Based on the pre-clinical data showing the significance of Zbtb20 in tumorigenesis, clinically oriented studies have been conducted on its diagnostic role [46]. As an immunohistochemical marker, Zbtb20 needs to adhere to several criteria to be considered of diagnostic significance.

### 5.1. Sensitivity

The biological processes affected by Zbtb20 occur in a broad set of tissues, with varying embryological origins. Zbtb20 expression is also susceptible to dysregulation due to viral DNA integration or inflammation. This makes Zbtb20 not very sensitive as a marker [22,66,67].

### 5.2. Specificity

Zbtb20 shows a strong specificity for malignant processes, which may make it a valuable marker in diagnostic pathology [68]. The best examples are found in liver pathology [46]. As a mitotically labile organ, the liver has an intermediate number of cells with low nuclear staining using an anti-Zbtb20 antibody. During liver regeneration, most of the cells have an intermediate nuclear reaction. However, malignant processes, most notably the classical variety of HCC, have strong nuclear staining in all tumor cells [46]. This makes Zbtb20 a valuable marker for suspected HCC in the preoperative biopsy, where only small tumor fragments are sent for histopathological evaluation. Such small pieces often exhibit only mild tissue atypia. Based on the limited number of cells available for analysis, especially in highly differentiated HCC, atypical cells may be difficult to distinguish from cells with a degenerative or inflammatory change. Using Zbtb20 as a marker may be essential in the differential diagnosis in such cases [68,69]. However, since Zbtb20 expression has been noted in several malignancies, a differential diagnosis between a primary and a metastatic liver tumor could not be possible based on the Zbtb20 expression alone. For example, one of the most common liver metastatic malignancies, GAC, also shows a nuclear reaction for an anti-Zbtb20 antibody [32,46,67]. In addition to HCC, Zbtb20 has also been proposed to have a high diagnostic value in the differential diagnosis between small-cell and non-small-cell carcinoma of the lung, most notably adenocarcinoma of the lung [26]. So far, the clinical application of Zbtb20 as a marker is still speculative and more data will be necessary.

### 5.3. Differential Expression

Anti-Zbtb20 immunostaining has a high affinity towards tumors of glandular origin, which proves extremely valuable in everyday pathology practice in highly differentiated adenocarcinomas (G1), where the differential diagnosis between a benign and malignant tumor is often difficult to place, especially on small biopsies and poorly differentiated ones (G3), where the histogenic group of the tumor can often be challenging to define [68,69]. Both of these conditions make Zbtb20 a prospective novel marker with potential clinical significance due to its utility in the proper diagnosis of glandular malignancies and the choice of correct treatment modalities for the patient [46,68].

## 6. Future Perspectives and Clinical Application (Discussion)

Genes regulating cell division, motility, survival, or inflammation are the cornerstones of cancer biology. Here we show that Zbtb20 modulates all these cellular processes. The bulk of the research concentrates on HCC. All data show that overexpression of Zbtb20 in tumor tissue is associated with poor patient prognosis. Furthermore, Zbtb20 is a target gene for HBV integration, resulting in its overproduction by hepatocytes. Zbtb20 overexpression in healthy mice livers leads to liver hypertrophy—a risk factor for cancer development. Even though the involvement of Zbtb20 in HCC is the best-studied case, several reports show that it is also overexpressed in many other malignant tumors (GAG, AML, NSCLC, GBM, and breast cancer). In this paper, we point out two possible clinical uses of Zbtb20. First, it may prove useful in the field of molecular pathology as a tool in the immunophenotypic analysis of tumors. Nonspecific markers for tumor differentiation are widely used to help the diagnostic procedure. For example, (1) proliferative capacity may be determined by Ki67, Cyclins (D1), and CDKIs like p15 and p21; (2) blood vessel growth by VEGF and CD31; (3) apoptotic activity can be defined by activated Caspase; and (4) tumor suppression activity by p53 [70]. These markers do not provide information on the molecular type of the studied malignancies. For differential diagnosis, the expression of a group of cell- or tissue-specific markers is generally used [66,71]. In our opinion, Zbtb20 may be of clinical significance in the differential diagnostics and staging of HCC. Similar to other markers, Zbtb20 would probably be helpful within a panel of markers because its individual interpretation may lead to misdiagnosis. This is mainly because some metastatic lesions, like GAC and NSCLC, also express Zbtb20 at lower levels [26,32]. Furthermore, research on CUX1 suggests that some HCC may be Zbtb20 -negative—an attractive prospective area of study [57]. This may lead to the coupling of Zbtb20 and CUX1 as markers, allowing for a more precise diagnosis and possibly a new clinical classification of HCC. Second, Zbtb20 may prove a useful tool in genetic screening—mutations of the Zbtb20 gene appear in several studies as a risk factor for esophageal and breast cancers. As already mentioned, Zbtb20 has an HVB integration site and is associated with more aggressive HCC [22]. Third, Zbtb20 was shown to induce an inflammatory response following TLR stimulation by LPS [64]. H. pylori LPS can induce inflammation by stimulating TLR2 [36,72]. High levels of Zbtb20 in patients with H. pylori infection would imply a stimulated NF-kB pathway. This risk factor is further exacerbated by the fact that Zbtb20 is a risk gene in GAC and is associated with higher migratory and proliferative capabilities of GAC cell lines [27,32,51,52].

## Figures and Tables

**Figure 1 ijms-24-13763-f001:**
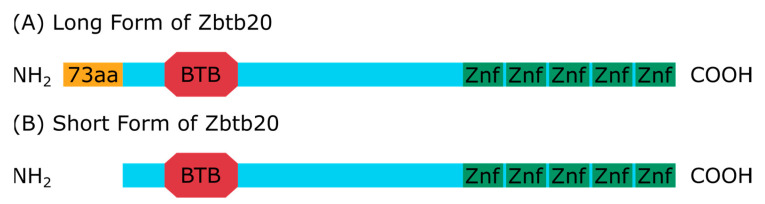
A schematic representation of the Zbtb20 gene and its encoded domains. (**A**,**B**) Both isoforms have a region encoding a BTB domain, followed by 5 zinc fingers. (**A**) The long form of Zbtb20 has an extra 73 amino acid residues at the NH2 terminus, giving it a total length of 741 amino acids. (**B**) The short isoform lacks these residues, which results in a protein length of 668 amino acids. Abbreviations: aa, amino acids; Znf, zinc finger.

**Figure 2 ijms-24-13763-f002:**
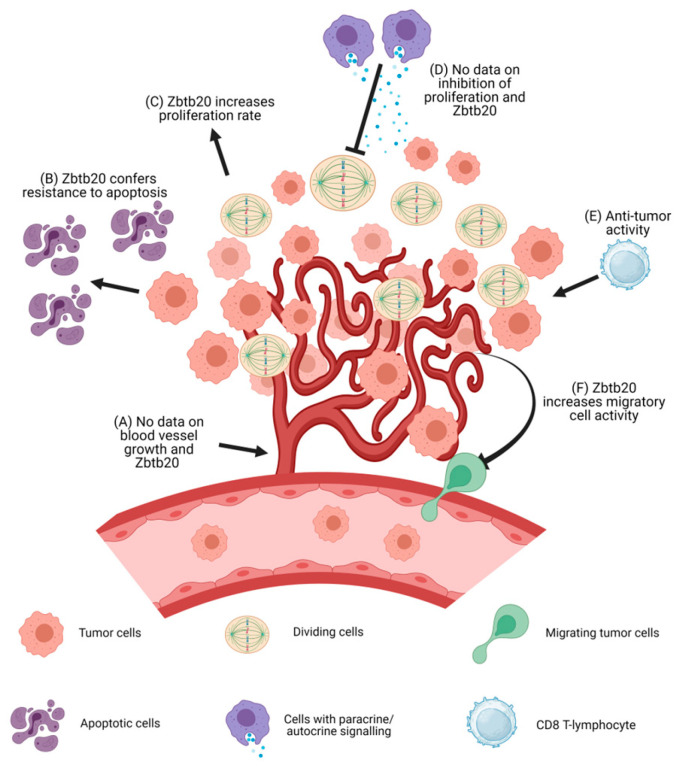
A schematic representation of tumor formation is shown above along with a summary of the known effects of Zbtb20 in the pathogenesis and development of malignant tumors. (**A**) No data are available on the effects of Zbtb20 on blood vessel growth. (**B**) Zbtb20 overexpression leads to lower levels of apoptosis in cell cultures. (**C**) Zbtb20 overexpression leads to increased proliferation rate. (**D**) It is unknown if Zbtb20 confers any resistance to proliferation inhibitors. (**E**) Zbtb20 knock-out increases the anti-tumor activity of CD8+ cytotoxic T cells. (**F**) Zbtb20 overexpression leads to an increase in the migratory activity of cells. Increases in the levels of MMPs have been observed. Abbreviations: MMPs, matrix metalloproteinases.

**Figure 3 ijms-24-13763-f003:**
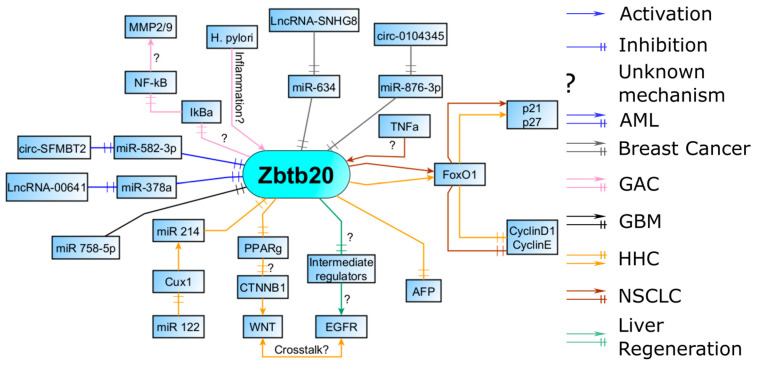
AML—Lnc RNAs, miRs, and circular RNAs regulate Zbtb20 levels in AML; breast cancers, similar to AML, are regulated by a myriad of non-coding RNAs. GAC—Zbtb20 expression is increased during inflammation and is involved in the induction of the proinflammatory NF-kB pathway. Increased activity and synthesis of MMPs, necessary for migration, is most likely a result of the NF-kB activation. GBM—miRs help suppress Zbtb20. HCC—regulatory pathway through FOXO1 is also present in HCC. An alternative HCC phenotype is possible through CUX1 signaling with concomitant Zbtb20 repression. NSCLC—Zbtb20 can influence proliferation through the FOXO1 pathway. Inflammation seems to upregulate Zbtb20 expression here as well. Liver regeneration—Zbtb20 facilitates EGFR induction which is found in some HHC. Abbreviations: AML, acute myeloid leukemia; GAC, gastric adenocarcinoma; GBM, glioblastoma multiforme; HCC, hepatocellular carcinoma; Lnc RNAs, long noncoding RNAs; NSCLC, non-small-cell lung carcinoma.

## Data Availability

No new data were created or analyzed in this study. Data sharing is not applicable to this article.

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
