# Peer review of "Transcription Factor Zbtb20 as a Regulator of Malignancy and Its Practical Applications"

_ijms, 2023, doi:10.3390/ijms241813763_

Round 1

Reviewer 1 Report

The review comprehensively summarized function of ZBTB20 in tumor malignance and progression. Few studies have been fully addressed on ZBTB20 role in tumor pathogenesis, so authors have make a lot as to better present ZBTB20 potential therapeutically targeting role. Few questions need to be discussed before acceptance.

1. ZBTB is a huge family that have dozens of proteins. I would suggest listing some ZBTB transcription factors, especially some have clear role in promoting tumors.

2. Some limitations in features of malignant tumors. Obviously these are key features but not all. tumor microenvironment is another aspect to be discussed.

3. Details need to be listed in figure 1. I dont understand presentation in the figure.

No comments.

Author Response

Reviewer One

  1. ZBTB is a huge family that have dozens of proteins. I would suggest listing some ZBTB transcription factors, especially some have clear role in promoting tumors.

Our Response: We did add some general information on ZFPs and for other ZBTB proteins in the introduction. The focus of this review is on one particular ZFP. Adding too much information for other members of this family of proteins will divert the attention and lead to a prohibitively long review. Also, we did add a suggestion of a recent excellent review of ZFP and cancer biology by Zhao et all in the introduction (The Role of Zinc Finger Proteins in Malignant Tumors. The FASEB Journal).

ADDED SECTION: “Zinc fingers are a large group of proteins which contain at least one of the following domains KRAB, BTB, SET and SCAN along with the zing fingers [1]. The importance of zing finger proteins in cancer biology was excellently reviewed by Zhao et all [2]. Several members of the ZBTB protein group are well established in the field of cancer biology both as tumor suppressors such as ZBTB16, ZBTB28, and oncogenes like ZBTB3, ZBTB7A, ZBTB38 and others [3–8].”

  1. Some limitations in features of malignant tumours. Obviously, these are key features but not all. tumor microenvironment is another aspect to be discussed.

Our Response: That is an excellent point! There is no specific information other than some possible effects of Zbtb20 on immune cells. Mostly articles are concerned with B-cell development. There is one work that links Zbtb20 in CD-8 cell and antitumor immunity (in short ZBTB20 KO CD8 have better antitumor activity). See https://doi.org/10.4049/jimmunol.2000459 for further information. We added a small section discussing that possibility titled “2.5Anti-tumour immunity”. Figure 2 (previously figure 1) was redacted to include this information. As we mentioned in the revies itself, effects of Zbtb20 are not studied in blood vessels biology. Same applies to fibroblast and matrix production, to our best knowledge.

ADDED SECTION “2.5 Anti-tumour immunity

Immune cells are a key feature of the tumor microenvironment. A recent study provided a glimpse into the role of ZBTB20 in tumor immunity and showed that deleting ZBTB20 enhances the anti-tumor activity of CD8+ cytotoxic T-cells.[38] This is quite interesting since ZBTB20 overexpression is usually linked to a more malignant tumor phenotype.”

  1. Details need to be listed in figure 1. I don’t understand presentation in the figure.

Our Response: Thank you for the suggestion! The figure really is a bit ambiguous. We did change the figure itself, and added some annotations so it is easier to understand. Explanation were written in the figure text. The figure is now Figure 2 (since one more figure was added due to a request from reviewer 2).

Reviewer 2 Report

The submitted manuscript entitled “Transcription factor Zbtb20: regulator of malignancy and practical applications” discusses the roles of a transcription factor ZBTB20 in malignant tumors. This manuscript scientifically sounds and may be of interest for the journal audience. The manuscript contains 60 references and 23 of them were published last 5 year. 2 Figures are presented to illustrate the results obtained. However, there are some concerns and recommendations to improve the quality of the manuscript. These are as follows:

1.     It is recommended to provide illustrative information regarding structural properties of this protein: its domains, motifs, etc. I think that this should be included because the manuscript is devoted to a certain protein and its functions.

2.     What are the relationships of ZFP120 with other proteins of ZFP family? See an excellent review article at https://doi.org/10.1038/cddiscovery.2017.71.

3.     Also, recently, interesting results were obtained on the orchestration of cell response to stress stimuli and cancer with the involvement of transcription factors including zinc finger proteins. See and discuss the next article at https://doi.org/10.3390/metabo12050464.

4.     Line 70: You write: “Higher protein levels of ZBTB20…”. How much protein are you talking about in this case? As a whole, there is no information about the Zbtb20 protein amounts in cells, tissues.

5.     Authors indicate that ZBTB20 is a direct repressor of AFP. Please, add special information about AFP as an important biomarker for HCC. The next article is recommended: https://doi.org/10.1080/14737159.2021.1987217.

6.     There are mistakes. For example, Figure 1. A summary of the know effects of Zbtb20; line 367. Third ZBTB20…

7.     English grammar should be improved. 

Should be improved

Author Response

Dear reviewer,

You will find attached our responses. 

Thank you for your time and comments,

Kind regards,

The authors

Round 2

Reviewer 2 Report

I am satisfied by the authors' corrections and responses

Minor editing is required